# Growth, Tolerance, and Compliance of Infants Fed an Extensively Hydrolyzed Infant Formula with Added 2′-FL Fucosyllactose (2′-FL) Human Milk Oligosaccharide

**DOI:** 10.3390/nu13010186

**Published:** 2021-01-09

**Authors:** Carlett Ramirez-Farias, Geraldine E. Baggs, Barbara J. Marriage

**Affiliations:** Scientific & Medical Affairs, Abbott Nutrition, Columbus, OH 43219, USA; carlett.ramirez@abbott.com (C.R.-F.); geraldine.baggs@abbott.com (G.E.B.)

**Keywords:** cow milk allergy, formula intolerance, hydrolyzed formula, 2′-FL, human milk oligosaccharide

## Abstract

Background: The purpose of this study was to evaluate the growth, tolerance and compliance effects of an extensively hydrolyzed formula with added 2′-FL in an intended use population of infants. Methods: A non-randomized, single-group, multicenter study was conducted. Infants (0–60 days of age) with suspected food protein allergy, persistent feeding intolerance, or presenting conditions where an extensively hydrolyzed formula (eHF) was deemed appropriate were enrolled in a 2-month feeding trial. The primary outcome was maintenance of weight for age z-score during the study. Weight, length, head circumference, formula intake, tolerance measures, clinical symptoms and questionnaires were collected. Forty-eight infants were enrolled and 36 completed the study. Results: Weight for age z-scores of infants showed a statistically significant improvement from study day 1 to study day 60 (0.32 ± 0.11, p = 0.0078). Conclusions: Overall, the results of the study demonstrate that the study formula was well tolerated, safe and supported growth in the intended population.

## 1. Introduction

Extensively hydrolyzed infant formulas (eHF) are often the first formula recommended for infants with suspected or confirmed cow’s milk allergy or protein sensitivity who are not breast-fed or who do not tolerate breast milk. They are also used in infants with persistent intolerance symptoms such as fussiness, diarrhea, spit-up, vomiting, or constipation.

Ongoing research has shown that human milk oligosaccharides (HMOs) play an important role in breast-fed infants. HMOs provide several benefits, such as acting as a prebiotic, acting as a soluble decoy receptor to prevent pathogen adhesion to the infant gut and enhancing the development of the immune system [1,2]. Mature human milk contains 5–15 g/L of oligosaccharides, representing the 3rd largest solid component, following lactose and fat, and is present at about a 20-fold higher concentration than that found in bovine milk [3,4,5]. Levels of HMOs vary between individuals and over the course of lactation [3,6,7,8]. The most abundant HMO in 80% of mother’s milk, namely that of secretor donors, is 2′-fucosyllactose (2′-FL), which ranges from 0.06 to 4.65 g/L [8].

In previous growth and tolerance studies, healthy infants have been fed formula containing 2′-FL. Overall, no safety concerns were observed, and growth and measurements of tolerance were not significantly different between the experimental formulas with added 2′-FL and control formula [9,10,11]. In addition, infants fed the formula containing 2′-FL had circulating cytokine concentrations that differed from the control fed infants but did not differ from the breast-fed infants [12].

The purpose of this study was to assess the growth, tolerance, compliance and change in clinical symptoms in infants with suspected food protein allergy or persistent feeding intolerance on an extensively hydrolyzed casein-based infant formula with added 2′-FL.

## 2. Materials and Methods

Between March of 2019 and January of 2020, 48 infants were enrolled in a prospective, single group, non-randomized multicenter study. The protocol was conducted in accordance with Good Clinical Practices.

This study was approved by the Institutional Review Board (IRB) and was registered at clinicaltrials.gov (NCT03884309). All participants’ parents gave written informed consent approved by an Independent Ethics Committee/IRB and signed Health Insurance Portability and Accountability Act (HIPAA) documents.

### 2.1. Study Population

Inclusion criteria included infants that were less than 60 days of age with a past medical history and clinical symptoms that accounted for the infant being placed on a hypoallergenic extensively hydrolyzed formula (eHF).

The infants were consuming an eHF formula without 2′-FL, either Alimentum^®^ (Abbott Nutrition, Columbus, OH, USA), or Nutramigen^®^ (Mead Johnson Nutritionals, Evansille, IN, USA) for persisting feeding intolerance symptoms, symptoms of suspected food protein allergy sensitivity or other conditions where eHF was deemed appropriate by their health care professional. Parent(s) of infants confirmed their intention not to administer prescription medications, over-the counter (OTC) medications (such as Mylicon^®^ for gas), home remedies (such as juice for constipation), prebiotics, probiotics, herbal preparations or rehydration fluids that might affect gastrointestinal (GI) tolerance, unless the infants were currently consuming and had been directed by their health care professional to continue their use during the study. Parent(s) also confirmed their intention not to administer solid foods, juices, vitamins and minerals (with the exception of vitamin D). Infants received the study formula ad libitum as their sole source of nutrition.

Infants were excluded if they were receiving oral or inhaled steroids or if they had been treated with antibiotics (with the exception of topical antibiotics), or other medications that may affect growth, GI tolerance and/or development, within 2 weeks prior to enrollment. Exclusion also included an adverse maternal, fetal or participant medical history that was thought by the investigator to have potential for effects on growth, and/or development or if the infant had an allergy or intolerance to any ingredient in the study product, as reported by the parent.

### 2.2. Methods

The study formula was a clinically labelled hypoallergenic casein-based powdered eHF designed to provide 20 kcal per fl oz at standard dilution with 0.2 g/L of 2′-FL. (Similac^®^ Alimentum^®^, Abbott Nutrition, Abbott Laboratories, Columbus, OH, USA). The study formula with the exception of the added 2′-FL was the Similac^®^ Alimentum^®^ commercially available in the US.

At Visit 1, eligible infants consuming an eHF without 2′-FL were switched to the study formula and enrolled into the 60 ± 5 day feeding trial. At SDAY 1 the mean ± SEM was 33.2 ± 2.8 days of age. Weight, length and head circumference measurements were collected and plotted on the 2006 World Health Organization (WHO) growth standards [13] at study entrance, Visit 3 (30 ± 3 days), and Visit 4 (60 ± 5 days). Intake and stool records were completed by parents and an Infant Feeding and Stooling Pattern Questionnaire was completed at Visit 1 to assess the baseline status of infants. Daily intake and stool records were recorded up to Visit 2 (study day (SDAY) 7), and 3 consecutive days before Visit 3 (SDAY 30 ± 3) and Visit 4 (SDAY 60 ± 5) to assess tolerance and change in perceived clinical status after feeding the study formula.

The primary outcome was maintenance of weight for age z-score during the study. The secondary variables were mean rank stool consistency (MRSC), predominant stool color, predominant stool consistency, average daily number of stools per day, percent of feedings with spit-up/vomit associated with feeding (within 1 h), weight, interval weight gain per day, length, interval length gain per day, head circumference (HC), and interval HC gain per day.

Supportive variables were average volume of study formula intake per day (mL/day), parental responses to the Infant Feeding and Stool Patterns Questionnaire, the Formula Satisfaction Questionnaire and Clinical Symptoms Questionnaire. Clinical symptoms reported by the parents were reviewed at enrollment, visit 2 (SDAY 7) and Visit 4 (SDAY60) by the investigators.

Subject demographics (birth weight, sex, race, ethnicity and age at enrollment) were collected. Safety monitoring consisted of the collection of adverse events and serious adverse events during the study.

### 2.3. Statistical Methods

A total sample size of 30 subjects has 80% power to detect a mean change of −0.206 in weight for age z-scores from study entry to exit, assuming a standard deviation (SD) from a previous study of 0.39, using a two-sided 5% level paired t-test. These mean difference and SD correspond to the 70th percentile of the normal distribution, that is, we expect about 70% of the infants to have a change in z-score of −0.206 or better. Based on an estimated 35% attrition rate, enrollment of 47 infants was planned. The nQuery^®^ Advisor 8.0 (“Statsols” (Statistical Solutions Ltd), Cork, Ireland) sample size software and SAS^®^ version 9.4 (SAS Institute Inc., Cary, NC, USA) were used. Data summaries are reported as Mean ± SEM for continuous data, and *n* (%) for categorical data.

## 3. Results

Forty-eight subjects were enrolled, but one never received product. Hence, 47 subjects were included in the intent-to-treat (ITT) data. Thirty-six (76.6%) of the ITT subjects were included in the protocol evaluable (PE) cohort. Eleven (23.4%) failed one or more evaluability criteria, including consumption of non-study feeding for more than 5 days (*n* = 2), use of medications that may affect GI tolerance (*n* = 1), SDAY 60 anthropometric measurement obtained outside the window (*n* = 1), premature discontinuation of study product (*n* = 6) and lost to follow-up (*n* = 1). The reasons for premature discontinuation of formula were as follows: Parent reported AE (*n* = 1), Investigator reported AE (*n* = 1), Parent requested discontinuation for reason other than AE (*n* = 2), Non-compliance (*n* = 1) and Lost to follow-up (*n* = 1). The 36 PE subjects, 22 male and 14 female, with gestational age of 38.6 ± 0.2 weeks, were 33.2 ± 2.8 days of age at enrollment, with a mean birth weight of 3262.5 ± 101.6 g. Other subject demographics (race, ethnicity, mode of delivery and pre-study feeding regimen) were also collected (see Table 1).

Weight z-scores of infants showed a statistically significant improvement from day 1 to SDAY 60 (Mean change ± standard error, SE: 0.32 ± 0.11, *p* = 0.0078). In addition, 78% (95% confidence interval: 61%, 90%) of the infants maintained their weight for age z-scores exceeding the hypothesized percentage of 70%. The evolution of weight for age z-score throughout the study is included in Table 2. Participant weight were below the 50th percentile throughout the study.

Infants fed eHF passed an average of 1.6 ± 0.1 stools per day at SDAY 1–7, 1.8 ± 0.2 at SDAY 28–30 and 1.7 ± 0.1 at SDAY 58–60. The average mean rank stool consistency (MRSC) was ranked by parents on a scale of 1 to 5 (1 = watery, 2 = loose/mushy, 3 = soft, 4 = formed, 5 = hard). MRSC at day 7, 30 and 60 was 2.41 ± 0.11, 2.55 ± 0.12, 2.47 ± 0.12, respectively. The predominant stool consistency was between loose/mushy to soft. The predominant stool color was brown.

The average volume of study formula intake per day was 754 ± 36 mL/day at SDAY 1–7, 850 ± 44 mL/day at SDAY 28–30 and 935 ± 44 mL/day at SDAY 58–60. A mean of 27.5 ± 4.4, 29.0 ± 4.8, 32.8 ± 5.0 percent of feedings had associated spit-up/vomit at SDAY 1–7, SDAY 28–30 and SDAY 58–60, respectively. Infants had a mean of 7.9 ± 0.2 study formula feedings per day at SDAY 1–7, 7.6 ± 0.2 at SDAY 28–30 and 7.1 ± 0.2 at SDAY 58–60.

At enrollment, even though the infants were currently consuming an extensively hydrolyzed formula they had the following persisting symptoms: diarrhea (*n* = 2), constipation (*n* = 6), blood in stool (*n* = 1), vomiting (*n* = 4), spit-up/gagging/reflux (*n* = 17), fussiness (*n* = 10) and rash or eczema (*n* = 7).

Clinical History and Symptoms were recorded after 7 and 60 days of switching from an extensively hydrolyzed formula without 2′-FL to eHF with 2′-FL. Parents of infants fed eHF without 2′-FL with persisting symptoms at entry reported after 7 days of switching to eHF with 2-’FL that: Eczema resolved or improved in 71% of infants, and in 29% remained the same. Vomiting resolved or improved in 100% of infants, constipation resolved or improved in 84% of infants, and spit-up/gagging/reflux improved in 59% of infants and in 35% remained the same. Fussiness improved or resolved in 40% and remained the same in 50% of infants. Diarrhea resolved in 50% and in 50% of infants remained the same. Blood in stool resolved in 100% of infants after day 7 (See Table 3).

After 60 days, parents reported: that eczema resolved or improved in 72% of infants, and in 29% remained the same. Vomiting resolved in 75% and remained the same in 25% of infants, constipation resolved or improved in 83% of infants and in 17% remained the same. Spit-up/gagging/reflux improved in 71% of infants and in 29% remained the same. Fussiness improved or resolved in 60% and remained the same in 40% of infants. Blood in stool stayed as resolved in 100% of infants after day 60 (See Table 3).

In addition, parents of infants fed eHF without 2′-FL with persisting symptoms at entry reported after 7 days of switching to eHF with 2′-FL that: Constipation worsened in 17% of infants, but after 60 days it was resolved.

Spit-up/gagging/reflux worsened in 6% of infants after 7 days, but after 60 days it resolved, and fussiness worsened in 10% of infants, but after 60 days it resolved. Overall, after 60 days of switching to eHF with 2′-FL all persisting symptoms either remained the same, improved or resolved. (See Table 3).

## 4. Discussion

Extensively hydrolyzed formulas are commonly used in infants with persistent intolerance symptoms, and in infants with suspected or documented cow’s milk allergy or protein sensitivity. Clinical experiences with infant formula supplementation with 2′-FL has found it to be safe, well-tolerated and support growth. The study formula is presently not available in the US, but a very similar formulation of the eHF with added 2′-FL is available in international markets. A recent clinical study assessing hypoallergenicity of an extensively hydrolyzed whey-based formula with 2′-FL and lacto-*N*-neotetraose (LNnT) demonstrated no issues with tolerance in the open feeding portion of the study [14]. In addition, preclinical studies have shown that 2′-FL can attenuate the allergic responses in a food allergy model, which presents a potential improvement in the nutritional management of infants with food allergies [15,16].

Our results are in agreement with other studies which reported normal growth patterns in healthy infants consuming eHF and adequate tolerance in infants with suspected food protein allergy or intolerance [17,18]. The addition of 2′-FL to the eHF was shown to be safe and well tolerated.

Tolerance measures such as MRSC, stool color, average volume of formula intake and percent of feedings associated with spit-up/vomit after 1 hr of feeding per day were comparable to other eHF feeding studies that did not contain 2′-FL. Interestingly, it also demonstrated parent reported improvement in symptoms in infants that were already on an eHF formula. A limitation of this study is the length of time the infants were consuming an eHF without 2′-FL before starting the study formula. Additional clinical research may reveal additional beneficial effects of 2′-FL in the allergic population.

Adverse events (AEs) were observed in 15 (32%) infants in the study (ITT cohort). Most AEs were mild in severity and deemed by the investigators as not related to product. The most common reported AEs were seborrheic dermatitis (five infants), gastrointestinal reflux disease (three infants) and infantile spit-up (two infants).

## 5. Conclusions

The results of this study demonstrate that eHF formula with added 2′-FL was well accepted enabling adequate volume to demonstrate a statistically significant improvement of weight for age z-scores. The formula was safe and well tolerated and consumption of the formula over 60 days showed improvement and resolution of persistent symptoms.

## Figures and Tables

**Table 1 nutrients-13-00186-t001:** Demographic data.

**Gender, *n* (%)**	
Male	22 (61.1)
Female	14 (38.9)
**Is Participant Hispanic/Latino, *n* (%)**	
Hispanic or Latino	3 (8.3)
Not Hispanic or Latino	33 (91.7)
**Mode of Delivery, *n* (%)**	
Vaginal	24 (66.7)
C-Section	12 (33.3)
**Pre-study Feeding Regimen, *n* (%)**	
Alimentum without 2′-FL	33 (91.7)
Nutramigen without 2′-FL	3 (8.3)
**Gestational Age: Weeks**	
Mean ± SEM	38.6 ± 0.2
**Age at SDAY 1: Days**	
Mean ± SEM	33.2 ± 2.8
**Birth Weight (g)**	
Mean ± SEM	3262.5 ± 101.6

**Table 2 nutrients-13-00186-t002:** Weight for Age Z-score (WHO).

	Visits		
	SDAY 1	SDAY 30	SDAY 60
Mean ± SEM	−0.37 ± 0.20	−0.24 ± 0.19	−0.05 ± 0.17
Median	−0.24	−0.12	0.04
Q1, Q3	−0.75, 0.28	−0.74, 0.59	−0.54, 0.83
Std Dev	1.18	1.11	1.02
Min, Max	−3.94, 1.62	−3.55, 1.52	−2.31, 2.04
*n*	36	35	36

Q1 = 25th percentile, Q3 = 75th percentile.

**Table 3 nutrients-13-00186-t003:** Change in Clinical Symptoms in Infants at 7 and 60 days after switching to eHF with 2′-FL.

Clinical Symptoms	Prior to Being Placed on a Hypoallergenic Extensively Hydrolyzed Formula	At Study Initiation	At Study Visit 2 (Day 7), n (% of Subjects with the Symptom at Entry)	At Study Visit 4 (Day 60), n (% of Subjects with the Symptom at Entry)
**Diarrhea**	9	2	1 (50%)—Same 1 (50%)—Resolved	2 (100%)—Better
**Constipation**	18	6	4 (67%)—Better 1 (17%)—Resolved 1 (17%)—Worse	1 (17%)—Same 3 (50%)—Better 2 (33%)—Resolved
**Blood in stool**	4	1	1 (100%)—Resolved	1 (100%)—Resolved
**Vomiting**	11	4	1 (25%)—Better 3 (75%)—Resolved	1 (25%)—Same 3 (75%)—Resolved
**Spit-up/Gagging/Reflux**	25	17	6 (35%)—Same 10 (59%)—Better 1 (6%)—Worse	5 (29%)—Same 10 (59%)—Better 2 (12%)—Resolved
**Fussiness**	24	10	5 (50%)—Same 2 (20%)—Better 2 (20%)—Resolved 1 (10%)—Worse	4 (40%)—Same 2 (20%)—Better 4 (40%)—Resolved
**Rash or Eczema**	9	7	2 (29%)—Same 4 (57%)—Better 1 (14%)—Resolved	2 (29%)—Same 3 (43%)—Better 2 (29%)—Resolved

## Data Availability

The data that support the findings of this study are available from the corresponding author (BJM) upon request.

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
