# Peer review of "Growth, Tolerance, and Compliance of Infants Fed an Extensively Hydrolyzed Infant Formula with Added 2′-FL Fucosyllactose (2′-FL) Human Milk Oligosaccharide"

_nutrients, 2021, doi:10.3390/nu13010186_

Round 1

Reviewer 1 Report

The authors studied the growth, tolerance and safety of the infants, who had the cow milk allergy, when they fed the hydrolyzed infant formula with added 2’-FL, showing that these were secured for them. The experiments were precisely designed without any ethical problem, and the data were logically concluded. I suggest that the authors should reply for the following comments.

  1. How many percent of 2’-FL was incorporated into the milk replacer? Is the test formula commercially available?
  2. It is likely that the intervention trial was performed using the infant formula without 2’-FL at first and then switched using that with 2’-FL, evaluating the growth, clinical findings and stool consistency. I suggest that the time schedule of the interventional trial is described in Methods.
  3. It is unlikely that the comparison of the growth, stool consistency and tolerance was performed between the subjects fed with 2’-FL and those without 2’-FL in this study. I suggest that the advanced issues for the infant growth and health by feeding 2’-FL may be discussed in the discussion.
  4. I suggest that the information of demographics (birth weight, sex, race, ethnicity, age at enrollment, mode of delivery and pre-study feeding regimen, etc) of the subjects will be shown in a table.

A minor point

Line 37

“namely that of Secretor donors” will be added after 80% of mother’s milk.

Author Response

Thank you for your review of the manuscript.

1.How many percent of 2’-FL was incorporated into the milk replacer? Is the test formula commercially available?

The amount of 2'-FL that was in the formula was 0.2 g per L. This has been added to the manuscript under methods (line 77).

The test formula is not commercially available in the US, but a very similar formulation of the extensively hydrolyzed  casein based formula with added 2'-FL is commercially available in international markets. Added a comment in the discussion on line 175.  Also added detail on the study formula in the Methods (line 77).

2. It is likely that the intervention trial was performed using the infant formula without 2’-FL at first and then switched using that with 2’-FL, evaluating the growth, clinical findings and stool consistency. I suggest that the time schedule of the interventional trial is described in Methods.

Added additional information on the time schedule of the intervention is included (line 81). At SDAY 1 the mean ± SEM was 33.2 ± 2.8 days of age. A limitation of the study was the length of time that the infants were on an extensively hydrolyzed casein formula without 2'-FL before starting the study formula with 2'FL. Addressed this limitation  in the discussion (line 188)

3. It is unlikely that the comparison of the growth, stool consistency and tolerance was performed between the subjects fed with 2’-FL and those without 2’-FL in this study. I suggest that the advanced issues for the infant growth and health by feeding 2’-FL may be discussed in the discussion.

Added additional information in the discussion as growth, stool consistency and tolerance were only assessed at baseline in the infants consuming the extensively hydrolyzed formula without 2'-FL. Added a reference  (# 14, Nowak-Wegrzyn et al) demonstrating that HMO added to an extensively hydrolyzed formula was well tolerated. Also added a paragraph discussing tolerance measure in other studies that did not contain 2'-FL. (line 185).

4. I suggest that the information of demographics (birth weight, sex, race, ethnicity, age at enrollment, mode of delivery and pre-study feeding regimen, etc) of the subjects will be shown in a table.

Excellent suggestion. Added Table 1 for demographics.

A minor point

Line 37

“namely that of Secretor donors” will be added after 80% of mother’s milk.

Thank you -comment added (line 37)

Reviewer 2 Report

This is a nice paper, overall well written.

Growth is the primary outcome. Therefore, the reviewer had hoped to see a table or figure on the weight-for-age z-line evolution (or another parameter) of these infants on the formula. The way the results are formulated, it could be interpret as excessive weight gain.

Methods: it should be specified that infants were switched from the "old" Alimentum to the "new" , from without to with HMO. The content of Similac Alimentum in HMO should be given.  

Premature discontinuation: what were the reasons?

Prior to inclusion: how long had the infants been on the "previous" eHF?

Would it be possible to perform statistics on the change in clinical symptoms?

References : missing reference to be discussed: Confirmed Hypoallergenicity of a Novel Whey-Based Extensively Hydrolyzed Infant Formula Containing Two Human Milk Oligosaccharides.  Nowak-Wegrzyn A, Czerkies L, Reyes K, Collins B, Heine RG. Nutrients. 2019 Jun 26;11(7):1447. doi: 10.3390/nu11071447.

Author Response

Thank you for your review

Growth is the primary outcome. Therefore, the reviewer had hoped to see a table or figure on the weight-for-age z-line evolution (or another parameter) of these infants on the formula. The way the results are formulated, it could be interpret as excessive weight gain.

Have included a table (Table 2) which shows the evolution of weight for age z-scores during the trial. Participants weights were below the 50th percentile (z-score =0) throughout the trial.

Methods: it should be specified that infants were switched from the "old" Alimentum to the "new" , from without to with HMO. The content of Similac Alimentum in HMO should be given.

In the demographic table we have included the pre-feeding regimen which included both commercially available Alimentum and Nutramigen. Added the amount of 2'-FL (0.2g/L) added to Alimentum (line 78).

Premature discontinuation: what were the reasons?

Added the reasons for premature discontinuation of formula in the results section (line 114)

Prior to inclusion: how long had the infants been on the "previous" eHF?

A limitation of the study was the length of time that the infants were on an extensively hydrolyzed casein formula without 2'-FL before starting the study formula with 2'FL.  The average age of enrollment was 33.2 ± 2.8 days, but the length of time on the previous extensively hydrolyzed formula was not recorded. Addressed this limitation in the discussion (line 179)

Would it be possible to perform statistics on the change in clinical symptoms?

It was considered not advisable to perform statistical test on the change of symptoms for the following reasons: 1) The appropriate test for the categorical data responses (resolved, better, same, or worse) is a chi-squared goodness of fit test. 2) However, one of the assumptions for a valid chi-square test, that the expected frequencies must be at least 5 in all cells, was not satisfied for most but one (Spit-up/Gagging/Reflux) due to small sample sizes.

For the exception Spit-up/Gagging/Reflux, the proportion of participants whose symptoms were better from entry to visit 2 (59%) and from entry to visit 4 (59%) was higher than the proportion in the other categories (p≤0.028).